# Zwitterionic Modification of Polyethyleneimine for Efficient *In Vitro* siRNA Delivery

**DOI:** 10.3390/ijms23095014

**Published:** 2022-04-30

**Authors:** Fengfan Liu, Huahui Su, Mengqian Li, Wanxuan Xie, Yunfeng Yan, Qi Shuai

**Affiliations:** 1National Engineering Research Center for Process Development of Active Pharmaceutical Ingredients, Collaborative Innovation Center of Yangtze River Delta Region Green Pharmaceuticals, Zhejiang University of Technology, Hangzhou 310014, China; liufengfan@zjut.edu.cn (F.L.); z742711352@Gmail.com (H.S.); limengqian_1006@163.com (M.L.); 2112123011@zjut.edu.cn (W.X.); 2College of Biotechnology and Bioengineering, Zhejiang University of Technology, Hangzhou 310014, China

**Keywords:** siRNA delivery, nanoparticle, polyethyleneimine, zwitterionic, anti-protein adsorption

## Abstract

Polyethylenimine (PEI) has been widely used in gene delivery. However, its high cytotoxicity and undesired non-specific protein adsorption hinder the overall delivery efficacy and the practical applications of PEI-based gene delivery systems. In this study, we prepared hydrophobically modified PEIs (H-PEIs) via the reaction of octanal with 40% of primary amines in PEI_25k_ and PEI_10k_, respectively. Two common zwitterionic molecules, 1,3-propanesultone and β-propiolactone, were then used for the modification of the resulting H-PEIs to construct polycationic gene carriers with zwitterionic properties (H-zPEIs). The siRNA delivery efficiency and cytotoxicity of these materials were evaluated in Hela-Luc and A549-Luc cell lines. Compared with their respective parental H-PEIs, different degrees of zwitterionic modification showed different effects in reducing cytotoxicity and delivery efficiency. All zwitterion-modified PEIs showed excellent siRNA binding capacity, reduced nonspecific protein adsorption, and enhanced stability upon nuclease degradation. It is concluded that zwitterionic molecular modification is an effective method to construct efficient vectors by preventing undesired interactions between polycationic carriers and biomacromolecules. It may offer insights into the modification of other cationic carriers of nucleic acid drugs.

## 1. Introduction

Gene therapy is a technique used to treat diseases by delivering the therapeutic genetic materials into cells. It has provided new promising options for the treatment of genetic disorders, cardiovascular diseases, and cancers [1,2,3]. Since the approval of the first gene therapy drug by the FDA in 1998, there has been a tremendous development in this area, and more than 20 gene therapies have been launched worldwide [4]. Clinical trial data in recent years have also demonstrated the effectiveness and safety of gene therapy for the treatment of a variety of human diseases [5,6]. More importantly, the rapid development of gene editing technology, such as TALENs, ZFNs and CRISPR-Cas [7,8], has further tapped the clinical potential of gene therapy. However, direct delivery of naked therapeutic genes is highly challenging due to the large molecular weight, intensive electronegativity, and the high hydrophilicity of genes [9]. Upon exposure to physiological conditions, intrinsic fragile genes undergo fast degradation within seconds, making it more difficult for gene therapy to achieve clinical outcomes [10]. Therefore, the development of safe and effective delivery systems is still of great significance for the clinical applications of gene therapeutics.

An ideal gene delivery system should be able to load enough nucleic acids (NAs), protect them effectively, and transfer them efficiently into target cells with minimal adverse effects [11]. Nonviral vectors, such as lipids/lipid-like materials and cationic polymers, are normally considered safe and are of high priority for the development of a gene delivery system [12,13,14,15]. They can bind nucleic acids efficiently via electrostatic interactions and form quite stable nanostructures, ensuring efficient delivery of genes in vivo [16]. Their physicochemical properties can be readily tuned by modifying chemical structures, making them suitable for the delivery of different types of nucleic acids [17,18]. In addition, they can be prepared on a large scale with affordable costs and high reproducibility, which is of great importance for the potential commercialization of nucleic acid-based drugs [19]. Among them, lipid-based gene delivery systems have made remarkable progress in both preclinical and clinical stages [20]. For example, ONPATTRO® (patisiran), the first RNAi drug approved by the FDA, is formulated as a lipid complex for the delivery of therapeutic small interfering RNA (siRNA) to hepatocytes [21]. However, lipid-based drug delivery systems also have some drawbacks, such as low efficiency caused by poor stability and rapid blood clearance, toxicity, immunogenicity, and poor targeting [22,23]. Cation polymers with versatile chemical structures also play important roles in gene delivery [24]. They effectively combine with genes through electrostatic interactions to form polyplex nanoparticles (NPs), which protect genes from degradation by extracellular nucleases and thus prolongs their circulation time in vivo. The enriched positive surface charge of cationic polymer-based NPs enhances endocytosis and promotes endosomal escape of genes via the “proton sponge effect” [23]. Notably, PEI has a prominent position among the most investigated cationic polymers for gene delivery. Its high density of cationic charge does allow for effective condensation with genes and significantly facilitates interaction with cellular membranes [25]. Alternately, it also leads to poor biocompatibility and high toxicity of PEI-based gene delivery systems [26].

In order to balance the delivery efficacy and biocompatibility, a lot of methods have been developed to modify PEI carriers [27,28,29]. Among them, hydrophobic modification of PEI has been proven effective to improve gene delivery with less cytotoxicity [30]. Generally, hydrophobic modification endows PEI/NAs NPs with superior features including improved stability via additional hydrophobic interactions, enhanced endocytosis and endosomal escape due to the presence of hydrophobic moieties [29], and reduced toxicity [31]. On the other hand, hydrophilic modification, such as PEGylation, has also been considered essential for PEI-based vectors to improve overall delivery efficacy. As an important alternative of PEGylation, zwitterion modification of vectors has demonstrated excellent ability in reducing non-specific protein adsorption by forming strong and stable hydration shells on gene polyplex NPs [32]. In addition, zwitterionic NPs are able to avoid the undesired long-term accumulation of NPs in reticuloendothelial system organs, making zwitterioncally modified PEI free of hazards [33]. In previous studies, the construction of polymers with zwitterions was mainly achieved through the polymerization of limited zwitterionic monomers, such as 3-[[2-(methacryloyloxy)ethyl]-dimethylammonio]propionate (CBMA) and 2-methacryloyloxyethyl phosphorylcholine (MPC) [34]. Zwitterionic modification can be a versatile strategy for the introduction of zwitterionic units to gene carriers. It is predicted that zwitterionic modification will afford PEI vehicles improved gene delivery efficacy and biocompatibility.

In this study, we constructed hydrophobically and zwitterionically modified PEI for the delivery of siRNA to cancer cells. Hydrophobically modified PEIs (H-PEIs) were synthesized by the reaction of octanal with 40% of primary amines in PEI_10k_ and PEI_25k_. Then, zwitterions were introduced to H-PEIs via the reaction of amines with 1,3-propanesultone and β-propiolactone, producing N-sulfopropylated H-PEI (H-PEIs-S) and N-carboxyethylated H-PEI (H-PEIs-C), respectively. The capability of these PEI derivatives in delivering siRNA was assessed in HeLa-Luc and A549-Luc cells. siRNA loaded NPs fabricated with selected zwitterionic PEIs were fully characterized. The results revealed that the zwitterionic modification of PEIs endowed NPs with reduced cytotoxicity and non-specific protein adsorption, enhanced endocytosis, and improved overall in vitro siRNA delivery efficiency.

## 2. Results and Discussion

### 2.1. Synthesis of Hydrophobically and Zwitterionically Modified PEI

Hydrophobic modification of PEI has been widely proved to improve overall gene delivery efficiency of PEI-based NPs by facilitating gene encapsulation and enhancing cellular internalization [29]. In this work, 40% of primary amines of PEI was first modified via reductive condensation with octanal. The resulting hydrophobically modified PEIs (H-PEIs) were further modified with 1,3-propanesultone or β-propiolactone through ring-opening addition, affording zwitterionic PEIs (H-zPEIs) with different modification degrees (Figure 1a). In total, 12 zwitterionic PEIs derivatized from PEI_25K_ and PEI_10K_ were prepared, and the results are summarized in Figure 1b. The formation and modification degree of H-PEI-S was determined by ^1^H NMR spectra (Figure 1c and Appendix A). The structure of H-PEI-C was verified by FT-IR spectra (Figure 1d), although the exact zwitterion modification degree was not available from ^1^H NMR due to the overlap of characteristic peaks with parent PEI.

### 2.2. siRNA Silencing and Cytotoxicity Evaluation on HeLa-Luc and A549-Luc Cells

With these modified H-zPEI libraries in hand, we next examined the effects of zwitterionic modification of PEI on the efficacy of siRNA delivery and identified efficient siRNA delivery carriers for further evaluation. Anti-luciferase siRNA used to knock down the luciferase gene was chosen as a model nucleic acid agent, and the gene silencing effects of H-zPEIs/siRNA NPs were assessed by luciferase assay in both HeLa-Luc and A549-Luc cells which stably express luciferase. Commercial reagent RNAiMax under its optimal conditions was chosen as control. In general, the overall siRNA delivery efficiency of these H-zPEIs/siRNA NPs in HeLa-Luc cells is highly related to the molecular weights of parent PEIs, zwitterion agents, and the degree of zwitterionic modification (Figure 2a). To our delight, when 40% of primary amines in PEI_25k_ were modified with n-octanal via reductive condensation, the resulting PEI_25k_ derivative (25–40) showed significantly reduced toxicity with only a slight sacrifice of gene silencing efficiency. Subsequent zwitterionic modification of 25–40, either with 1,3-propanesultone or β-propiolactone, further reduced the cytotoxicity of H-zPEIs, although the knockdown of luciferase expression induced by these H-zPEIs/siRNA NPs was also gradually reduced. This is mainly attributed to the reduction in positive charge in PEIs due to the introduction of negative zwitterionic moieties. Similar trends upon hydrophobic and zwitterionic modification were also observed for PEI_10k_, although the overall siRNA delivery efficiency of all these PEI_10k_ derivatives was poorer than their PEI_25k_ counterparts. Finally, desired H-zPEI-based siRNA carriers with high gene silencing efficiency (>70%) and almost 100% cell viability were screened out, including 25-40-S-10, 25-40-S-20, 25-40-C-20, 25-40-C-30, and 25-40-C-40 (Figure 2a).

In our investigation on siRNA delivery to A549-Luc cells, it was true that the gene silencing efficiency was poorer for almost all carriers, but siRNA NPs prepared with 25-40-S-10, 25-40-S-20, 25-40-C-20 and 25-40-C-30 still achieved similar results as the control RNAiMax (Figure 2b).

To further validate the siRNA delivery efficacy of the above modified PEI, 25-40-S-20/siRNA NPs and 25-40-C-30/siRNA, NPs were selected for dose response experiments in both HeLa-Luc cells (Figure 3a) and A549-Luc cells (Figure 3b), respectively. Generally, the efficiency of gene silencing was positively correlated with the concentration of siLuc. In HeLa-Luc cells, both 25-40-S-20/siRNA NPs and 25-40-C-30/siRNA, NPs exhibited gene silencing at doses as low as 8.6 nM siRNA (25 ng/well of NPs) concentration, whereas only at 17.1 nM siRNA (50 ng/well of NPs) concentration did these NPs show gene silencing in A549-Luc cells. When the concentration was further increased to 68.4 nM, significant cytotoxicity was observed for these NPs in both cell lines.

### 2.3. Characterizations of siRNA NPs Based on H-zPEIs

As control, when complexed with siRNA, PEI_25k_ modified with octanal (25–40) were able to form nanoparticles with a suitable size and positive surface charge (Figure 4a). As we expected, further zwitterionic modification of H-PEIs with either 1, 3-propanesultone or β-propiolactone afforded NPs with larger hydrodynamic diameters based on DLS measurements. The hydrodynamic diameter of siRNA NPs with *N*-sulfopropylated H-PEI (25-40-S-20) was in particular slightly larger than those with *N*-carboxyethylated H-PEI (25-40-C-30), which might be attributed to the stronger hydration of sulfonate in water. In addition, increased positive potentials were observed for NPs with 25-40-S-20 and 25-40-C-30, which are beneficial for gene delivery via enhanced effective interactions between siRNA-loaded NPs and cell membranes. In addition, because of the higher degree of zwitterionic modification, *N*-carboxyethylated H-PEI (25-40-C-30) had lower zeta potential. RiboGreen assays showed that siRNA could be completely complexed with all three modified PEI carriers when the polymer/siRNA mass ratio was 30:1 (Figure 4b). According to the polymer/siRNA mass ratio designed for the above in vitro gene transfection experiments, the consumption of primary amines in PEI induced by modification with octanal and zwitterion agents did not affect its siRNA binding capacity at the polymer/siRNA mass ratio designed for the above in vitro gene transfection experiments. The spherical shapes of these three types of nanoparticles were verified by TEM (Figure 4c). The apparent sizes of NPs were similar to those measured by DLS.

Introduction of zwitterionic fragments to PEI is expected to enhance the stability of H-zPEIs/siRNA NPs under serum conditions. High hydrophilicity of zwitterionic moieties has demonstrated its ability to alleviate nonspecific protein adsorption. Bovine serum albumin (BSA) adsorption was examined on the selected H-zPEIs/siRNA NPs with PEI_25k_-based NPs as control (Figure 5a). Obviously, the strongest BSA adsorption was detected for PEI/siRNA NPs, which adsorbed more than 95% of free BSA in solution. After being modified with octanal, the resulting H-PEIs (25-40) showed remarkably reduced BSA adsorption capacity. Only 50% of free protein was adsorbed by 25-40 siRNA NPs, which was mainly attributed to the octanyl modification which increased the hydrophobicity of PEI, and partially shielded the positive surface charge of NPs. Further reduction of BSA adsorption to 25% was observed for both 25-40-S-20 and 25-40-C-30 siRNA NPs with higher zeta potentials (Figure 4a). As expected, the increased surface charge induced by zwitterionic modification did not result in undesired protein adsorption, which might be explained by the fact that the strong hydration of zwitterionic moieties on these NPs played an important role in reducing BSA adsorption [32].

In addition, effects of zwitterionic modification on the stability of nanoparticles was also evaluated by incubating NPs with 50 % FBS, which simulated the systematic circulation conditions and would lead to nuclease degradation induced by nuclease (Figure 5b). Compared with naked siRNA that degraded within 1 h upon incubation, the content of siRNA in 25-40-S-20 NPs and 25-40-C-30 NPs showed little change in 6 h, indicating that these zwitterionic modifications on PEI could potentially protect siRNA payloads from nuclease degradation under physiological conditions.

### 2.4. Cellular Internalization of NPs Prepared with Selected Carriers

Cellular uptake of NPs is one of the crucial processes for efficient siRNA delivery and successful gene transfection. We next evaluated the cellular uptake of the selected NPs to further explore the effect of zwitterionic modification on siRNA delivery in vitro. As shown in Figure 6a, all of Cy5-siRNA loaded 25-40 NPs, 25-40-S-20 NPs and 25-40-C-30, NPs were endocytosed into Hela-Luc cells within 2 h. There was little difference in the Cy5 signal between 2 h incubation and 6 h incubation, suggesting that fast cellular uptake of all these NPs occurred. Quantitated data from flow cytometry showed similar results. Compared with 25-40/siRNA NPs, higher fluorescence intensity was observed for HeLa-Luc cells treated with 25-40-S-20/siRNA NPs and 25-40-C-30/siRNA NPs (Figure 6b,c), which might be due to the higher zeta potentials of these zwitterionic modified NPs. It is worth noting that the higher endocytosis achieved by zwitterionic modification does not lead to higher overall gene delivery efficiency, which suggests that other parameters including NPs stability, endosomal escape, and intracellular release may also play important roles in gene silencing for these siRNA loaded zwitterionic PEI-based NPs.

## 3. Experimental Section

### 3.1. Cell Culture

Human cervical cancer cells (HeLa-Luc) and human non-small cell lung cancer cells (A549-Luc) which stably express luciferase were cultivated in DMEM supplemented with 5% FBS and 1% penicillin/streptomycin. All cells were incubated at 37 °C in a humidified 5% CO_2_ atmosphere (Appendix A).

### 3.2. Materials

Polyethyleneimine (PEI, M_w_ = 25 kDa and 10 kDa), Dulbecco’s modified Eagle medium (DMEM), penicillin-streptomycin, fetal bovine serum (FBS), phosphate buffered saline (PBS), and 3-[4,5-dimethylthiazol-2-yl]2,5-diphenylterazolium bromide (MTT) were supplied by Sigma-Aldrich (Shanghai, China). 4′,6-Diamidino-2-phenylindole (DAPI) and Quant-iT RiboGreen RNA assay kits were acquired from Life Technologies (Carlsbad, CA, USA). The ONE-Glo + Tox luciferase assay kit was purchased from Promega (Madison, WI, USA). Octanal obtained from TCI (Shanghai, China). 1,3-propanesultone was purchased from Energy Chemical (Shanghai, China), and β-propiolactone from Adamas (Shanghai, China). Dialysis membrane was received from YOBIOS (Xi’an, China). All other reagents were purchased from Sigma-Aldrich (Shanghai, China). All chemicals were used as received without any further purification. Custom-synthesized siRNA against Luciferase (siLuc) (sense: 5′-GAUUAUGUCCGGUUAUGUA[dT][dT]-3′; antisense: 5′-UACAUAACCGGACAUAAUC[dT][dT]-3′), and Cy5-labeled siRNA (Cy5-siRNA) (sense: 5′-GAUUAUGUCCGGUUAUGUA[dT][dT]-Cy5-3′; antisense: 5′-UACAUAACCGGACAUAAUC[dT][dT]-3′) were purchased from Gene Pharma (Suzhou, China).

### 3.3. Preparation and Characterization of Zwitterionic PEI

Synthesis of hydrocarbon-modified PEI (H-PEI)**:** A solution of PEI (430 mg, 10 mmol) in methanol (5 mL) was added dropwise to a solution of octanal (513 mg, 4 mmol) in methanol (10 mL) while stirring. The mixture was continuously stirred for 12 h at room temperature, followed by the slow addition of 5 mL sodium cyanoborohydride (252 mg, 12 mmol) of methanol solution. The mixture was stirred for another 12 h at room temperature. After reaction, the solution was dialyzed against methanol for 3 days and the final octanyl modified PEI (H-PEI) was obtained via lyophilization. The structure of H-PEI was characterized by ^1^H NMR (Mercury pLUS-400, Varian Agilent Technologies, Palo Alto, CA, USA).

Synthesis of zwitterionic H-PEIs (H-PEI-S and H-PEI-C): 1,3-propanesultone solution in methanol (1 mol/L) was added dropwise to a solution of H-PEI (94.3 mg, 1 mmol) in anhydrous methanol (2 mL) under nitrogen atmosphere. The reaction was continuously stirred at 45 ℃ for 24 h. After that, the solution was dialyzed against ultrapure water for 3 days. After lyophilization, *N*-sulfopropylated H-PEI (H-PEI-S) was obtained as yellowish oil. *N*-carboxyethylated H-PEI (H-PEI-C) was prepared in a similar way by using β-propiolactone as zwitterionic modification agent. The structure of H-PEIs was characterized by ^1^H NMR (Mercury pLUS-400, Varian Agilent Technologies, Palo Alto, CA, USA) and FT-IR (Nicolet 6700, Thermo Fisher Scientific, Waltham, MA, USA).

### 3.4. Protein Adsorption Assay

In all, 100 μL of NP solution (1 mg/mL) and 100 μL of bovine serum albumin (BSA) standard solution (4 mg/mL) were mixed and then shaken at 37 °C for 0.5 h. The mixture was centrifuged at 10,000 rpm for 20 min and the supernatant was collected. The BSA concentration in the supernatants was determined using bicinchoninic acid (BCA) assay with a standard BSA calibration curve. The protein adsorption value A is defined as:(1)$$A=1-\frac{C_{s}V_{s}}{C_{i}V_{i}}$$
where C_i_ is the initial BSA concentration (4 mg/mL), C_s_ is the BSA concentration in the supernatant determined by BCA assay, V_i_ is the initial volume of the BSA standard solution (100 μL), and V_s_ is the total volume of the mixture (200 μL).

### 3.5. Preparation and Characterization of siRNA Polyplex NPs

siRNA polyplex NPs for in vitro studies were prepared with a polymer/siRNA ratio of 30:1 (wt/wt). Briefly, a solution of polymer in methanol was mixed with siRNA in deionized water by pipetting up and down 60 times. The resulting mixture was then left for 30 min at room temperature. Particle size and zeta potential of NPs were measured by dynamic light scattering method with a Zetasizer (Malvern Nano-ZS90, Malvern, UK) at 25 °C. The morphology of NPs was analyzed via transmission electron microscopy with a Hitachi HT-7700 instrument (Hitachi, Tokyo, Japan) operating at 120 kV.

### 3.6. siRNA Binding Assay

The siRNA binding capacity of NPs was tested by RiboGreen intercalation assay. RiboGreen working solution was prepared by diluting RiboGreen stock solution to 2000 times in 0.01M PBS buffer. To each well of 96-well plate were added 100 μL of polyplex NPs solution containing 20 ng siRNA and 100 μL of RiboGreen working solution, respectively. The fluorescence was measured at λ_ex_ = 485 nm and λ_em_ = 535 nm using hybrid multi-mode reader (BioTek Synergy H1). The amount of siRNA binding to polymers was calculated according to the standard curve established between free siRNA under a concentration gradient and respective fluorescence intensity [35].

### 3.7. Stability and Agarose Gel Retardation Assay

The stability of polyplex was examined by electrophoresis. H-zPEIs and siLuc RNA were mixed at polymer/siRNA mass ratio of 30:1, and then the resulting 5 μL polyplex NPs solution was mixed with equal volume of FBS. The resulting mixture was incubated at 37 °C for 0, 0.5, 1, 3, 6 h. After incubation, 1 μL of 100 nM EDTA solution was added, followed by another incubation in water bath (75 °C) for 10 min to deactivate nuclease. siRNA residuals were released from NPs by treating siRNA polyplex NPs with 5 μL SDS (40 g/L) of solution. A 5 μL prepared sample was moved by electrophoresis in agarose gel containing 2% ethidium bromide (1×TAE buffer, 100V, 15min). The siRNA bands on the gels were visualized and photographed using a Gel Imager System (BLT Gelview 6000Plus).

### 3.8. In Vitro Gene Silencing

The luciferase silencing experiments were performed in HeLa-Luc and A549-Luc cells. Cells were seeded into opaque white 96-well plates (10,000 cells/well) and incubated for 12 h. Then, the medium was replaced with 200 μL fresh medium, and 20 μL of nanoparticles (100 ng siRNA/well) was added to the wells, with RNAiMax as positive control. After incubation for 24 h, the media was gently removed. The cell viability and luciferase expressions were measured using ONE-Glo + Tox luciferase assay kits and normalized to untreated cells.

### 3.9. Cellular Uptake

The cellular uptake of NPs in HeLa-Luc cells was quantified on flow cytometry. Briefly, cells were seeded into transparent 12-well plates (300,000 cells/well) and incubated for 12 h. After exchanging the fresh medium, the cells were incubated with 100 μL NPs prepared using the Cy5-labeled siRNA for 6 h. Then the cells were collected, washed by PBS, and subjected to flow cytometry analysis. The mean fluorescence intensity of Cy5-positive cells was recorded.

The Cy5-labeled siRNA intracellular distribution was detected using fluorescence microscope. HeLa-Luc cells were seeded into transparent 96-well plates (10,000 cells/well) and incubated for 12 h. After exchanging the fresh medium, the cells were incubated with 20 μL NPs prepared using the Cy5-labeled siRNA for 2 and 6 h, respectively. Then, the cells were washed with PBS, fixed with 4% paraformaldehyde, and their nucleus was stained with DAPI. The images were captured using EVOS M7000 Cell Imaging System (Thermo Fisher, Waltham, MA, USA).

## 4. Conclusions

In summary, a small library of zwitterion-modified PEIs (H-zPEIs) was established by grafting 1,3-propanesultone or β-propiolactone in different ratios onto hydrophobically modified PEI_25k_ and PEI_10k_ via ring-opening addition reactions. Screening of gene delivery efficiency of H-zPEIs/siRNA NPs in Hela-Luc and A549-Luc cell lines indicated that introduction of zwitterions could reduce the cytotoxicity of PEI-based NPs, but gene transfection efficiency was also negatively affected. A good balance between transfection effect and cytotoxicity was achieved by both siRNA NPs prepared 25-40-S-20 and 25-40-C-30, respectively. Characterizations of siRNA loaded NPs fabricated with selected PEI derivatives revealed that all of them showed excellent siRNA binding capacity, reduced nonspecific protein adsorption, and enhanced stability upon nuclease degradation. Hydrophobic modification with octanal was crucial for the formation of stable NPs with appropriate sizes and zeta potentials. Further modification with zwitterion agents endowed H-zPEIs with superior anti-protein adsorption capacity, and the ability of resistance to protein adsorption increased with the increase in the proportion of zwitterion fragments. Cellular uptake experiments validated that zwitterionic modification could improve endocytosis of H-zPEIs/siRNA NPs. It is well demonstrated that zwitterionic modification is a potential strategy to enable polycationic carriers with good biocompatibility, long-term circulation, and an excellent gene silencing effect.

## Figures and Tables

**Figure 1 ijms-23-05014-f001:**
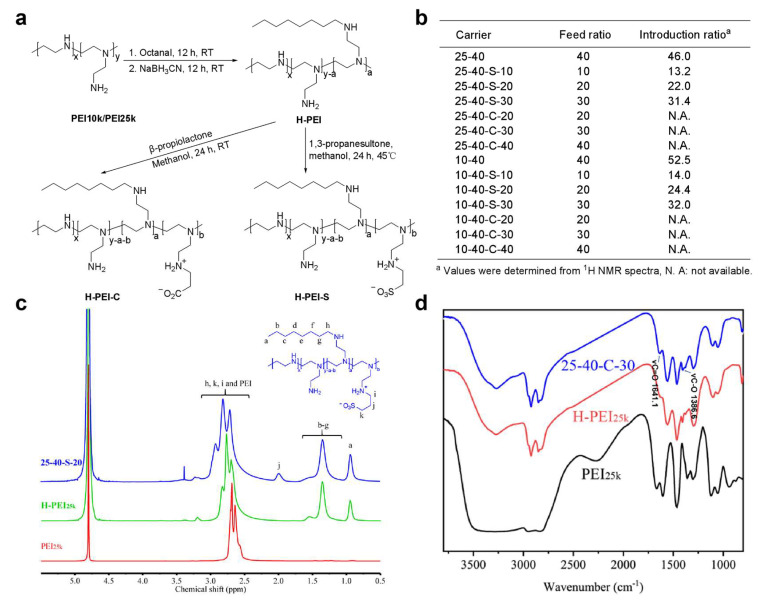
(**a**) Synthesis of H-zPEIs. (**b**) H-zPEIs with different zwitterion modification degrees. Final modified PEIs were named by the Mw of PEI, the octanylation degree, the zwitterion modification type (S for *N*-sulfopropylated and C for *N*-carboxyethylated), and zwitterion modification degree. (**c**) ^1^H NMR spectra of PEI_25k_, H-PEI_25k_, and H-S-PEI_25k_. (**d**) FT-IR spectra of PEI_25k_, H-PEI_25k_, and H-C-PEI_25k_.

**Figure 2 ijms-23-05014-f002:**
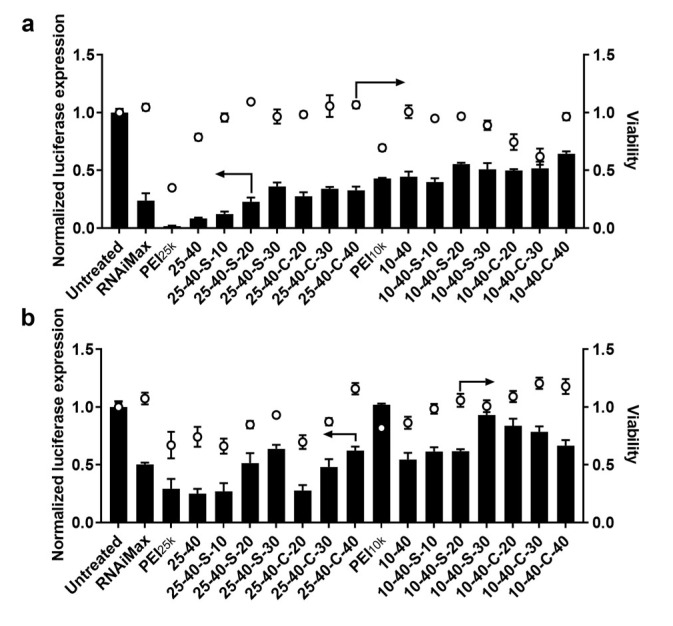
siRNA delivery efficiency of the H-zPEIs in (**a**) HeLa-Luc cells and (**b**) A549-Luc cells at polymer/siRNA mass ratio of 30:1 (34.2 nM siLuc). Dots represent cell viability and bars indicate relative luciferase expression compared to untreated cells (mean ± SD, n = 3).

**Figure 3 ijms-23-05014-f003:**
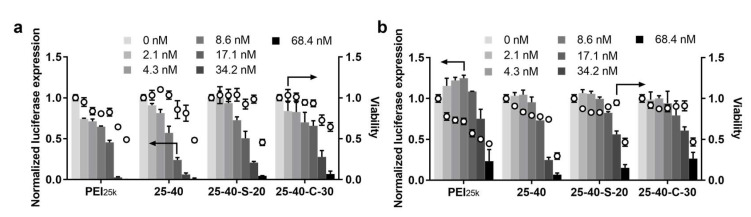
siRNA dose response for the top performing carriers. (**a**) HeLa-Luc and (**b**) A549-Luc. (mean ± SD, n = 3).

**Figure 4 ijms-23-05014-f004:**
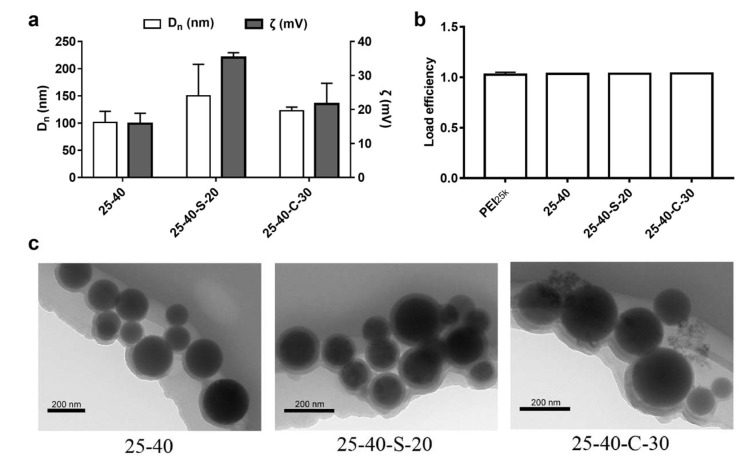
Characterizations of H-zPEI/siRNA NPs prepared at a polymer/siRNA mass ratio of 30:1. (**a**) Hydrodynamic diameters and zeta potentials of selected H-zPEI/siRNA NPs. (**b**) siRNA binding in selected NPs. (**c**) TEM images of selected NPs.

**Figure 5 ijms-23-05014-f005:**
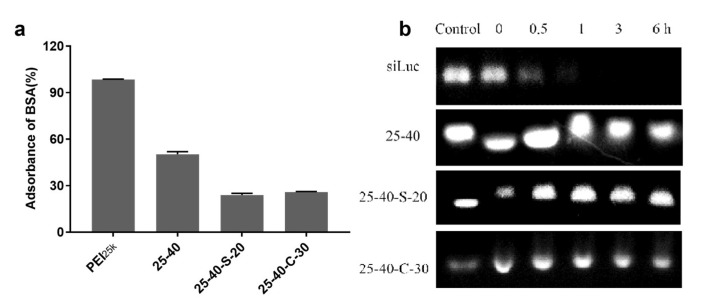
(**a**) Protein adsorption on selected H-zPEIs/siRNA NPs. (**b**) Stability of siRNA in selected H-zPEIs/siRNA NPs in 50% (v/v) fetal bovine serum by agarose gel retardation assay.

**Figure 6 ijms-23-05014-f006:**
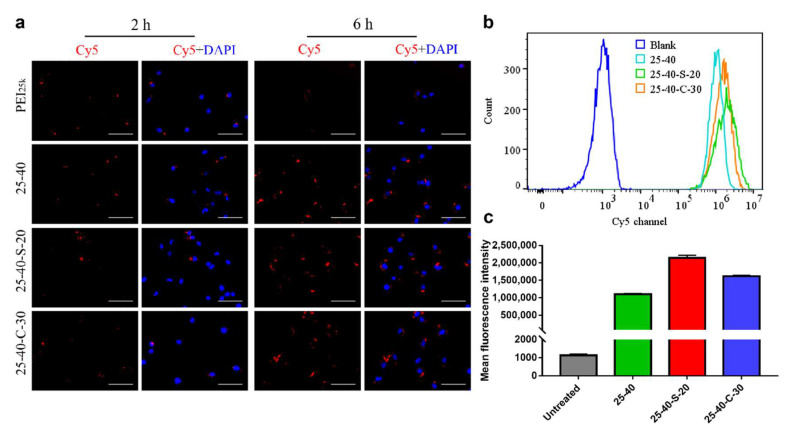
(**a**) Fluorescence images showed the intracellular distribution of H-zPEIs/Cy5-siRNA NPs. Scale bar: 75 μm (**b**) and (**c**) Cellular uptake of H-zPEIs/Cy5-siRNA NPs in HeLa-Luc cells determined by flow cytometry 4 h post-incubation.

## Data Availability

The data presented in this study are available on request from the corresponding author.

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
