# Peer review of "Zwitterionic Modification of Polyethyleneimine for Efficient In Vitro siRNA Delivery"

_ijms, 2022, doi:10.3390/ijms23095014_

Round 1

Reviewer 1 Report

Dear authors

In the manuscript, PEI polymer was modified with zwitterionic molecules to increase efficacy and reduce toxicity of siRNA. This strategy could potentially improve the RNAi approach for therapeutic purposes. There are several points to be clarified.

Major points

Was it demonstrated that the NPs have a zwitterionic characteristics? The authors should test NPs under different experimental conditions e.g pH

Is the synthesis of NP reproducible?

Figure 2: What are the arrows? Which is the negative control of the polymer?

A normal cell line must be tested.

Figure 3: What are the arrows? Are the values normalized to the untreated samples?

Figure 4: TEM images show a polydisperse NP that do not correspond to the DLS analysis.

Figure 6: scale bar value is not reported.

Which is the subcellular localization of the NPs?

Minors

English should be improved. Examples are:

…of human diseases with established safety (profile).

With these modified (the) H-zPEIs library in hand…

…whereas only at 17.1 nM siRNA (50 ng/well of NPs) concentration (did) these NPs show gene silencing in A549-Luc 153 Cells.

Please, convert BAS to BSA

Deioned..deionized

Author Response

Point 1: Was it demonstrated that the NPs have a zwitterionic characteristics? The authors should test NPs under different experimental conditions e.g pH

Response 1: Thanks for pointing out the issue. The authors did introduce Betaine-like structure to endow PEI with zwitterionic properties. Based on the results from NMR characterization of zwitterionically modified PEI (H-zPEIs) and DLS measurements of the corresponding NPs, it is confirmed that the resulting siRNA-loaded NPs based on H-zPEIs do possess characteristic zwitterionic properties.

Point 2: Is the synthesis of NP reproducible?

Response 2: Thanks for the question. Each test in this study has been repeated several times, and the process for the preparation of the NPs was repeatable.

Point 3: Figure 2: What are the arrows? Which is the negative control of the polymer?

Response 3: Thanks for pointing out the mistake. Figure 2 has been revised. Arrows were drawn to indicate that the bars correspond to the left Y-axis and circular symbols to the right Y-axis. The cells not treated with any NPs (untreated cells) is used as the negative control.

Point 4: A normal cell line must be tested.

Response 4: In this study, the gene silencing efficiency needs to be tested in gene-edited HeLa-Luc or A549-Luc cells by the reduction of luciferase level, while normal cell line without gene editing does not meet the requirements of the experiment.

Point 5: Figure 3: What are the arrows? Are the values normalized to the untreated samples?

Response 5: Thanks for the question. Arrows have been marked in more appropriate locations to avoid any possible confusion. The values were normalized to the untreated cells.

Point 6: TEM images show a polydisperse NP that do not correspond to the DLS analysis.

Response 6: Thanks for pointing out the issue. Inconsistencies between DLS measurements and TEM images appear to be a common problem. (J. Controlled Release 2022, 341, 616-633). It can be attributed to the differences in the environments to which the nanoparticles are exposed during DLS and TEM testing. NPs are measured in solution for DLS, while in air for TEM.

Point 7: Figure 6: scale bar value is not reported.

Response 7: Thanks for pointing out the mistake. The scale bar has been indicated in the caption of Figure 6.

Point 8: Which is the subcellular localization of the NPs?

Response 8: Thanks for the question. We used cy5-labeled siLuc RNA and polymers to prepare nanoparticles and the subcellular localization of the NPs is showed in Figure 6 by the cy5 signal (red).

Point 9: English should be improved. Examples are:

…of human diseases with established safety (profile).

With these modified (the) H-zPEIs library in hand…

…whereas only at 17.1 nM siRNA (50 ng/well of NPs) concentration (did) these NPs show gene silencing in A549-Luc 153 Cells.

Please, convert BAS to BSA

Response 9: Thanks for the kind suggestions. The manuscript has been thoroughly checked and revised. English and grammar have been improved.

Reviewer 2 Report

The paper focusses on the investigation of some non-viral vectors containing PEI and zwitterionic building blocks, mainly stressing the improvements brought by the zwitterionic unit on the performance of non-viral vectors.

In my opinion the design of the non-viral vectors is good and the methodology to investigate their performance was well chosen. The findings were not sound, but even so they can inspire other researchers. Some suggestions for the paper improvement are as follows.

  1. The use of zwitterionic units for building non-viral vectors is not new, and it is preferable in the Introduction to better highlight the achievements already reached. Overall, the choice of this design should be better justified.
  2. The sample codes on Figure 1 should be bolded to be easy seen by readers.
  3. The NMR spectra will be carefully revisited regarding the proton attribution. From figure 1c it appeared that the integral ratio of protons a/b-g does not correspond to the 3/12 value of the targeted structure.
  4. The pathway for calculation the modification degrees via NMR will be provided in Experimental part.
  5. Pay attention, different structures for H-PEI-S were given in Figure 1, in synthesis vs. NMR.
  6. It is not clear how the siRNA binding was determined. This will be clarified.
  7. The results of gel permeation assay will be better highlighted.

Author Response

Point 1: The use of zwitterionic units for building non-viral vectors is not new, and it is preferable in the Introduction to better highlight the achievements already reached. Overall, the choice of this design should be better justified

Response 1: Thanks for pointing out the issue. Serval non-viral vectors with zwitterionic units have been reported. However, in previous studies, the construction of polymers with zwitterionic units was mainly achieved by the polymerization of limited zwitterionic monomers. In this study, a more versatile strategy of zwitterion modification was used to construct vectors for siRNA delivery. The choice of this design has been justified in the revised manuscript.

Point 2: The sample codes on Figure 1 should be bolded to be easy seen by readers.

Response 2: Thanks for pointing out the issue. The sample codes have been bolded in Figure 1.

Point 3: The NMR spectra will be carefully revisited regarding the proton attribution. From figure 1c it appeared that the integral ratio of protons a/b-g does not correspond to the 3/12 value of the targeted structure

Response 3: Thanks for pointing out the issue. The more detailed NMR spectra data have been added to support information.

Point 4: The pathway for calculation the modification degrees via NMR will be provided in Experimental part.

Response 4: Thanks for the kind suggestion. The pathway for the calculation of the modification degrees via NMR has been added to support information.

Point 5: Pay attention, different structures for H-PEI-S were given in Figure 1, in synthesis vs. NMR.

Response 5: Thanks for the kind suggestion. The more detailed chemical structures of H-PEI-S and the corresponding instructions have been added to support information.

Point 6: It is not clear how the siRNA binding was determined. This will be clarified.

Response 6: Thanks for the kind suggestion. The Quant-iT RiboGreen RNA assay kits is a reagent that exhibits >1000-fold fluorescence enhancement when bound to RNA. Thus, free RNA in solution can be quantified by measuring the intensity of its fluorescence. (Anal. Biochem. 1998, 265, 368-374; Biomaterials, 2017, 118, 84-93.)

Point 7: The results of gel permeation assay will be better highlighted.

Response 7: Thanks for the kind suggestion. The nucleic acid gel permeation assay visually demonstrates the protective effect of vectors on RNA in the presence of 50% FBS. This result has been more appropriately described and explained in the revised manuscript.

Reviewer 3 Report

Comments:

PEI polymer bears primary, secondary, and tertiary amine groups which protonated at low pH, forming a polycation and condensing with siRNA or plasmid DNA into stable polyplexes. However, due to its multi-cation nature, mainly branched PEI shows high cytotoxicity. The authors have tried to synthesize hydrophobically modified PEI25k and PEI10k (H-PEIs) by reacting with octanal. Furthermore, they were modified with the zwitterionic molecules, 1,3-propanesultone and β-propiolactone to construct polycationic gene carriers. The authors tried to characterize the synthesized NP and tried to check some feasibility using in vitro study. Although the paper's subject is good, there are several major concerns that must be addressed before further consideration:

  1. Most of the polyplex (polymer/gene complex) are synthesized at the low pH value of 4~5 using citrate or acetate buffers. However, in this manuscript, the authors dissolved PEI or PEI modified polymer in methanol and then mixed it with siRNA in deionized water. How PEI or modified PEI will bear cation at neutral pH of distilled water and complex with negatively charged siRNA?
  2. The authors synthesized siRNA polyplex NPs using a polymer/siRNA ratio of 30:1 (wt/wt). The authors must discuss why they choose this ratio, is there any optimization done to select this ratio? There are no purification methods in the synthesis protocols, how are free siRNA or uncondensed siRNA removed?
  3. The authors claimed that, when the concentration of siRNA was increased to 68.4 nM, significant cytotoxicity was observed for these NPs in both cell lines. Is this cytotoxicity due to polymer or siRNA concentration?
  4.  As shown in Figure 2, the luciferase expression was very low in PEI treated HeLa-Luc and A549-Luc cells, in comparison to the other treated materials. Is this result due to better transfection effects of PEI than the modified PEI or RNAiMax or due to cytotoxicity of PEI (in which most of the cells expressing luciferase are dead)?
  5. Although the author claimed there is a better improvement of PEI cytotoxicity after modification, it is not significant as shown in Figure 2. In addition, modified PEI shows less transfection efficacy in comparison to the parental PEI which does not support the authors' conclusion in the abstract, “indicating that zwitterionic molecular modification is an effective method to improve siRNA delivery efficiency “
  6. On page 9, the agarose gel retardation and stability assay must be clearly stated in the different subtitles. If not, the authors must include a stability assay in the subtitle (i.e., Stability and agarose gel retardation assay). In the current format, this protocol is very confusing and lacks clarity. Similarly, Figure 5b caption must also include agarose gel retardation and stability assay. What is the importance of treating polyplex with SDS and how SDS will rupture polyplex to release siRNA?  
  7. The degree of grafting (%) octanal, 1,3-propanesultone and β-propiolactone on the PEI is not clear, needs more clarification in the main manuscript on how they did the calculation.

Author Response

Point 1: Most of the polyplex (polymer/gene complex) are synthesized at the low pH value of 4~5 using citrate or acetate buffers. However, in this manuscript, the authors dissolved PEI or PEI modified polymer in methanol and then mixed it with siRNA in deionized water. How PEI or modified PEI will bear cation at neutral pH of distilled water and complex with negatively charged siRNA?

Response 1: Thanks for the question. Acidic conditions favor the protonation of amino groups of materials. In fact, we did try to prepare polymer/siRNA complexes in citric acid-disodium hydrogen phosphate at pH 4.2. However, DLS measurement revealed that almost none of the resulting polyplex NPs had appropriate particle sizes and PDI, which is not good for further evaluation. In this study, optional neutral buffer and water turned out to be ideal conditions for polyplex preparation, by which smaller particle sizes and uniform distribution was achieved for some materials. (React. Funct. Polym. 2013, 73, 993-100; J. Appl. Polym. Sci.2021, 138, 51323).

Point 2: The authors synthesized siRNA polyplex NPs using a polymer/siRNA ratio of 30:1 (wt/wt). The authors must discuss why they choose this ratio, is there any optimization done to select this ratio? There are no purification methods in the synthesis protocols, how are free siRNA or uncondensed siRNA removed?

Response 2: Thanks for pointing out the issue. The mass ratio of 30:1 was chosen to prepare complexes for complete loading of siRNA. (Biomaterials, 2017, 118, 84-93; Proc. Natl. Acad. Sci. U. S. A., 2016, 113), E5702-E5710.) We did try to optimize the mass ratio and the 30:1 ratio was the best choice in this study. As shown in the siRNA binding assay, almost 100% of the siRNA was bound to the material and the obtained polyplex NPs were used without further purification. Actually, under our neutral aqueous conditions, the highly hydrophilic free siRNA is easily washed out during the preparation process.

Point 3: The authors claimed that, when the concentration of siRNA was increased to 68.4 nM, significant cytotoxicity was observed for these NPs in both cell lines. Is this cytotoxicity due to polymer or siRNA concentration?

Response 3: Thanks for the question. The administered dose was determined by the siRNA concentration in the nanoparticle solution. And thus, higher dosage of siRNA means higher polymer concentration which generally resulted in increased cytotoxicity.

Point 4: As shown in Figure 2, the luciferase expression was very low in PEI treated HeLa-Luc and A549-Luc cells, in comparison to the other treated materials. Is this result due to better transfection effects of PEI than the modified PEI or RNAiMax or due to cytotoxicity of PEI (in which most of the cells expressing luciferase are dead)?

Response 4: Thanks for the question. First, the transfection efficiency of each material has been normalized by the corresponding cytotoxicity. Secondly, PEI25k did show better gene knockout efficiency but it is also very toxic. In this study, zwitterionic modification of PEI successfully reduced its cytotoxicity without significant decrease of gene knockout efficiency, achieving a good balance between gene delivery efficacy and biocompatibility of PEI.

Point 5: Although the author claimed there is a better improvement of PEI cytotoxicity after modification, it is not significant as shown in Figure 2. In addition, modified PEI shows less transfection efficacy in comparison to the parental PEI which does not support the authors' conclusion in the abstract, “indicating that zwitterionic molecular modification is an effective method to improve siRNA delivery efficiency “

Response 5: Thanks for pointing out the mistake. Normally, it is difficult to construct a PEI derivative-based vector with higher transfection efficiency than parent PEI. In this study, a good balance was achieved between toxicity and transfection efficiency of the material by zwitterion modification. In addition, the conclusion has been revised.

Point 6: On page 9, the agarose gel retardation and stability assay must be clearly stated in the different subtitles. If not, the authors must include a stability assay in the subtitle (i.e., Stability and agarose gel retardation assay). In the current format, this protocol is very confusing and lacks clarity. Similarly, Figure 5b caption must also include agarose gel retardation and stability assay. What is the importance of treating polyplex with SDS and how SDS will rupture polyplex to release siRNA?

Response 6: Thanks for the kind suggestion. The subtitle of “agarose gel retardation assay” has been changed to “Stability and agarose gel retardation assay”, and more information has been added to the caption of Figure 5. Sodium dodecyl sulfate (SDS) is an anionic surfactant, which ruptures polyplex by competing with RNA for cationic polymer binding. (ACS Nano 2021, 15, 4576−4593)

Point 7: The degree of grafting (%) octanal, 1,3-propanesultone and β-propiolactone on the PEI is not clear, needs more clarification in the main manuscript on how they did the calculation.

Response 7: Thanks for the kind suggestion. The detailed NMR spectra data and the pathway for the calculation of modification degrees via NMR have been added to support information.

Reviewer 4 Report

Reviewer report on Manuscript Draft ‘Colorimetric Microneedle Patches for Multiplexed Transdermal Detection of Metabolites’

In this research, authors have developed a small library of zwitterion-modified poly-15 ethylenimine (H-zPEIs) was established 332 by grafting 1,3-propanesultone or β-propiolactone in different ratios onto hydrophobically modified PEI25k and PEI10k via ring-opening addition reactions. Screening of gene delivery efficiency of H-zPEIs/siRNA NPs in Hela-Luc and A549-Luc cell lines indicated that introduction of zwitterions could reduce the cytotoxicity of PEI-based NPs, but gene 336 transfection efficiency was also negatively affected. A good balance between transfection effect and cytotoxicity was achieved by both siRNA NPs prepared 25-40-S-20 and 25-40-C-30 respectively. Cellular uptake experiments validated that zwitterionic modification could improve endocytosis  of H-zPEIs/siRNA NPs. It is well demonstrated that zwitterionic modification is a potential strategy to enable polycationic carriers with good biocompatibility, long-term circulation and excellent gene silencing effect. The results revealed that the zwitterion modification of the PEI could efficiently reduce toxicity and non-specific protein adsorption, enhance endocytosis, and improve overall in vitro siRNA delivery efficiency.

In this research presented investigation is interesting, from the point of view of bioanalytical chemistry. The research is in scope of the journal. Therefore, the manuscript can be published after some minor corrections and improvements:

Authors report, that reported results revealed that the zwitterion modification of the PEI could efficiently reduce toxicity and non-specific protein adsorption, enhance endocytosis, and improve overall in vitro siRNA delivery efficiency. Therefore, cyto-compatibility of some other polymeric particles (Some Biocompatibility Aspects of Conducting Polymer Polypyrrole Evaluated with Bone Marrow-Derived Stem Cells. Colloids and Surfaces A: Physicochemical and Engineering Aspects 2014, 442, 152-156. // Evaluation of Cytotoxicity of Polypyrrole Nanoparticles Synthesized by Oxidative Polymerization. Journal of Hazardous materials 2013 250–251, 167–174.) could be overviewed and discussed. Recent reviews, that at some extent overviews an application nanopartricles for RNA delivery (Towards application of CRISPR-Cas12a in the design of modern viral DNA detection tools (Review). Journal of Nanobiotechnology 2022, 20, 41.) could be overviewed and discussed.

Data in Figures 6C, could be supported by error bars (if experiment was performed not once?) in order to estimate deviation of experimental results.

Author Response

Point 1: Authors report, that reported results revealed that the zwitterion modification of the PEI could efficiently reduce toxicity and non-specific protein adsorption, enhance endocytosis, and improve overall in vitro siRNA delivery efficiency. Therefore, cyto-compatibility of some other polymeric particles (Some Biocompatibility Aspects of Conducting Polymer Polypyrrole Evaluated with Bone Marrow-Derived Stem Cells. Colloids and Surfaces A: Physicochemical and Engineering Aspects 2014, 442, 152-156. // Evaluation of Cytotoxicity of Polypyrrole Nanoparticles Synthesized by Oxidative Polymerization. Journal of Hazardous materials 2013 250–251, 167–174.) could be overviewed and discussed. Recent reviews, that at some extent overviews an application nanopartricles for RNA delivery (Towards application of CRISPR-Cas12a in the design of modern viral DNA detection tools (Review). Journal of Nanobiotechnology 2022, 20, 41.) could be overviewed and discussed.

Response 1: Thanks for the kind suggestion. Polypyrrole has shown great potential as a drug carrier responsive to electrical stimulation. Excellent works have been done in the characterization of the morphology and the evaluation of related biological properties of modified polypyrroles. However, the construction of gene vectors based on polypyrrole has rarely been reported. The mentioned review of application nanopartricles for RNA delivery has been added to the manuscript and cited.

Point 2: Data in Figures 6C, could be supported by error bars (if experiment was performed not once?) in order to estimate deviation of experimental results

Response 2: Thanks for pointing out the mistake. The error bars have been added to Figure 6C by repeating the experiment three times.

Round 2

Reviewer 3 Report

The authors have tried to respond to each question, and almost addressed my concern in the current revised manuscript. I recommend this manuscript can be accepted and published after minor revisions.